# A note on precision-preserving compression of scientific data

Rostislav Kouznetsov[1,2]

[1]Finnish Meteorological Institute, Helsinki, Finland
[2]Obukhov Institute for Atmospheric Physics, Moscow, Russia

**Correspondence:** Rostislav Kouznetsov (Rostislav.Kouznetsov@fmi.fi)

**Abstract.** Lossy compression of scientific data arrays is a powerful tool to save network bandwidth and storage space. Properly applied lossy compression can reduce the size of a dataset by orders of magnitude while keeping all essential information, whereas a wrong choice of lossy compression parameters leads to the loss of valuable data. An important class of lossy compression methods is so-called precision-preserving compression, that guarantees that a certain precision of each number will be kept. The paper considers statistical properties of several precision-preserving compression methods implemented in "NetCDF operators" (NCO), a popular tool for handling and transformation of numerical data in NetCDF format. We compare artifacts resulting from use of precision-preserving compression of floating-point data arrays. In particular, we show that a popular Bit Grooming algorithm (default in NCO until recently) has sub-optimal accuracy and produces substantial artifacts in multipoint statistics. We suggest a simple implementation of two algorithms that are free from these artifacts and have double the precision. One of them can be used to rectify the data already processed with Bit Grooming.

We compare precision-trimming for relative and absolute precision to a popular Linear Packing (LP) method, and find out that LP has no advantage over precision-trimming at a given maximum absolute error. We give examples when LP leads to an unconstrained error in the integral characteristic of a field, or leads to unphysical values.

We analyse compression efficiency as a function of target precision for two synthetic datasets, and discuss precision needed in several atmospheric fields.

Mantissa rounding has been contributed to NCO mainstream as a replacement for Bit Grooming. The supplementary material contains the implementation of the algorithm in Python 3.

## 1 Introduction

Resolutions and the level of details of processes simulated with geoscientific models increase together with the increase of computing power available. Corresponding increase of the size of datasets needed to drive the models, and the datasets produced by the models makes the problem of transferring and archiving the data more and more acute.

The data are usually produced and stored as floating-point numbers implemented in most of computer systems according to the IEEE 754-1985 standard (ANSI/IEEE, 1985). The standard offers two formats: single-precision and double-precision. The data in these formats have precisions of approximately 7 and 16 significant decimal figures, and occupy 32 and 64 bits per value in a memory or on a disk.

The problem of storing and transferring the data, is normally addressed in three directions: increase of storage capacities and network bandwidth, reduction of the number of stored/transferred values by careful selection of needed variables and a wise choice of the archiving policies, and by applying various data compression techniques. We will focus on the latter ones.

Lossless compression algorithms map a data array to a smaller size in a way that the original data can be restored exactly. The algorithms are efficient for datasets of low information entropy rate, e.g. those that have a lot of repeating patterns.

Many scientific datasets have only a few (or even a few tens of) percent accuracy and thus require much less than 7 decimals to represent a value. When represented with standard formats, such datasets have seemingly-random numbers at less significant places, i.e. have high information entropy. As a result, application of lossless compression algorithms to such datasets does not lead to a substantial reduction of the dataset size. Transformations of a data array reducing its information entropy while introducing acceptable distortions, pose the basis for lossy compression algorithms. A transformation can make the array smaller, or can facilitate the performance of subsequent lossless compression. Note that the meaning of "acceptable distortion" depends on the data and/or the intended application.

An often used method of lossy compression is Linear Packing, when the original floating-point data are mapped to a shorter-length integer data by a linear transformation. The method, introduced in for GRIB (GRidded Binary) format approved back in 1985 by the World Meteorological Organization, and intended for use for gridded weather data (Stackpole, 1994), and is still in use in subsequent standards. Similar technique has been applied for NetCDF format according to the CF conventions (http://cfconventions.org). The Linear Packing into a shorter datatype allows for reduction of the storage space even without involving subsequent data compression. The method has been working well for the data of a limited dynamic range (within 1–3 orders of magnitude), however it causes poor representation for the data that have larger dynamic range or too skewed distribution of values.

Another class of lossy compression methods, so called, precision-preserving compression, is designed to guarantee a certain *relative* precision of the data. Setting a certain number of least-significant bits of the floating-point numbers in a data array to a prescribed value (trimming the precision) substantially reduces the entropy of the data making lossless compression algorithms much more efficient. A combination of precision-trimming and a lossless compression constitutes an efficient method of lossy compression with well controlled loss of relative precision. Unlike packing, a precision-trimming maps an array of floating-point numbers, to an array of floating-point numbers that do not need any special treatment before using.

Zender (2016) implemented precision-trimming in a versatile data-processing tool-set called "NetCDF operators" (NCO http://nco.sourceforge.net, last accessed on December 7, 2020), enabling the internal data compression features of the NetCDF4 format (https://www.unidata.ucar.edu/software/netcdf last accessed on December 7, 2020) to work efficiently. The tool-set allows a user to chose the needed number of decimals for each variable in a file, so the precision-trimming can be applied flexibly depending on the nature of the data. It was quickly noticed that a simple implementation of precision-trimming by setting non-significant bits to zero (bit shaving) introduces undesirable bias to the data. As a way to cope with that bias Zender (2016) introduced a Bit Grooming algorithm that alternately shaves (to zero) and sets (to one) the least significant bits of consecutive values. A detailed review of the compression methods used for scientific data as well as the analysis of

the influence of the compression parameters on compression performance can be found in the paper by Zender (2016) and references therein.

While comparing spatial structure functions of stratospheric ozone mixing ratio from satellite retrievals to those obtained from the Silam chemistry-transport model (http://silam.fmi.fi), and compressed with NCO, we have spotted a substantial distortion introduced by Bit Grooming into the structure function of the modelled fields. In the current note we investigate the cause of this distortions. We consider several precision-trimming algorithms and the inaccuracies they introduce. We suggest an implementation of mantissa-rounding that provides double the accuracy of the data representation than Bit Grooming with the same number of data bits, and does not introduce major artifacts in two-point statistics. In addition, we suggest a method of recovering the Bit-Groomed data.

The paper organized as follows. The next section formulates the concept of precision trimming and describes various methods for it and their implementation. Sec. 3 analyses the effect of precision-trimming methods on the introduced errors. Sec. 4 illustrates the effect of precision-trimming on statistics of two synthetic datasets. Besides relative-precision trimming we consider the properties of the absolute-precision trimming (Sec. 5) and Linear Packing (Sec. 6). The effect of trimming and Linear Packing on the storage size reduction is analysed in Sec. 7. Parameters of precision trimming suitable for several variables from atmospheric modelling are discussed in Sec 8. Conclusions are summarized at the end of the paper. Appendix contains example implementations of subroutines for precision-trimming.

## 2   Precision-trimming methods

Hereafter we will consider a single-precision floating-point representation, although all the conclusions can be extrapolated to arbitrary-precision numbers. A single-precision number according to ANSI/IEEE (1985) consists of 32 bits: 1 bit for sign, 8 bits for exponent and 23 bits for mantissa (also called "fraction"), having the most-significant bit (MSB) of mantissa implicit 1, therefore allowing for 24-bit resolution of mantissa:

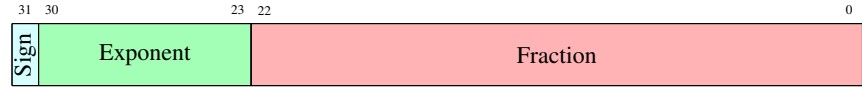

$M$ bits of mantissa allow for distinguishing any two numbers $a$ and $b$ if

$$2\frac{|a-b|}{|a|+|b|} > 2^{-M}, \tag{1}$$

therefore the representation provides $2^{-24} \simeq 6 \cdot 10^{-8}$ relative precision, or about 7 decimals.

If the data have smaller precision, the least-significant bits (LSBs) of mantissa contain seemingly random information, which makes lossless-compression algorithms inefficient. If one trims the precision, i.e. replaces those bits with a pre-defined value, the overall entropy of the dataset reduces, and the efficiency of compression improves. Therefore the problem of trimming LSBs can be formulated as follows: given an array of floating-point numbers set transform it so that the resulting values have all but N LSBs of their mantissa set to a prescribed combination, while the difference from the original array is minimal.

Hereafter we will use a term "keep-bits" for the N most-significant bits of mantissa, which we use to store the information from the original values, and "tail-bits" for the remaining bits of a mantissa that we are going to set to a prescribed value. The resulting precision is characterized by the number of bits of mantissa that we are going to keep.

The following methods have been considered:

- *shave* – set all tail-bits to zero

- *set* – set all tail-bits to one

- *groom* – set all tail-bits to ones or to zero in alternate order for every next element of the array

- *halfshave* – set all tail-bits to zero except for the most significant one of them, which gets set to one

- *round* – round mantissa to the nearest value that has zero tail-bits. Note that this rounding affects keep-bits, and might affect the exponent as well

The former three of them were implemented in NCO some years ago and are described well by Zender (2016). They can be implemented by applying of bit masks to the original floating-point numbers. *Halfshave* and *round* have been recently introduced into NCO. *Halfshave* is trivial to implement by applying an integer bit-mask to a floating-point number: first one applies the *shave* bit-mask and then sets MSB of tail-bits to 1.

Rounding can be implemented with integer operations (Milan Klöwer, private communication). Consider an unsigned integer that has all bits zero, except for the most significant one of the tail-bits. Adding this integer to the original value $u$, treated as unsigned integer would not change keep bits if $u$ has zero in the corresponding bit, and bit-shaving of the sum will produce the same result as bit-shaving the original value, i.e. $u$ rounded towards zero. If $u$ has one in the corresponding bit, the carry bit will propagate to the fraction or even to the exponent. Bit-shaving the resulting value is equivalent to the rounding $u$ towards infinity. In both cases the result is half-to-infinity rounding, i.e. rounding to the nearest discretisation level, and the value that is exactly in the middle of a discretisation interval would be rounded away from zero.

Rounding half-to-infinity introduces a small bias away from zero in average. The magnitude of the bias is half of the value of the least-significant bit of the original number, i.e. about $10^{-8}$ for the single-precision. For many applications such biases can be safely neglected. However, if one applies such a procedure in a sequence trimming the tail-bits one by one, the result will be equivalent to round to infinity, i.e. will have an average bias of the value of the most-significant of the tail-bits. A rigorous procedure of round half to even (a default rounding mode in IEEE 754) is free from biases and should be used.

The implementations of the above-mentioned bit-trimming and rounding procedures are given in Appendix.

## 3 Quantification of errors

To characterize the trimming we will use the number of explicit bits kept in the mantissa of a floating point number. Therefore "bits = 1" means that the mantissa keeps 1 implicit and 1 explicit bits for the value, and the remaining 22 bits have been reset to a prescribed combination.

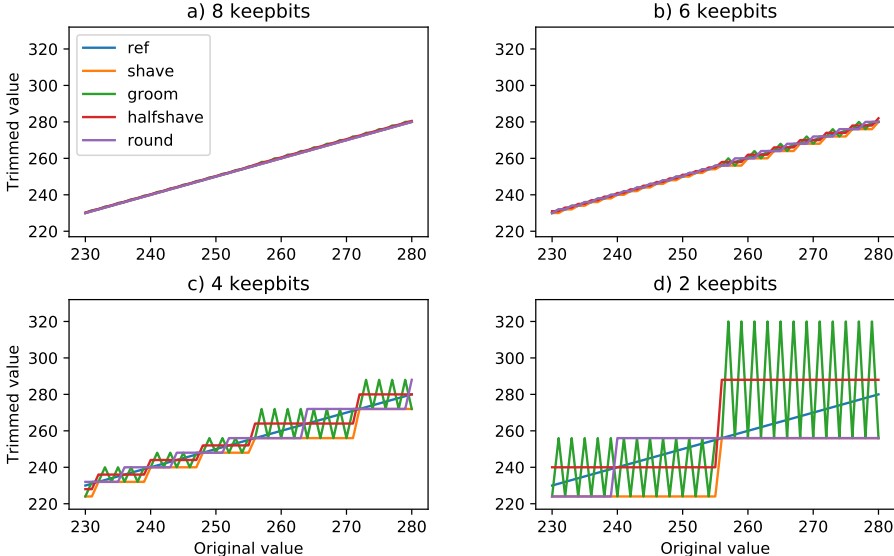

**Figure 1.** Trimmed values as a function of original values when trimmed to different number of keep-bits. The lines correspond to the different trimming methods.

When rounding to a fixed quantum (e.g. to integer), the magnitude of maximal introduced error is a function of the quantum. The error is one quantum for one-sided rounding (up, down, towards zero, away from zero), and half of the quantum for rounding to the nearest value. With precision-preserving compression, the quantum depends on the value itself. For a value $v$ discretized with $M$ bits of mantissa kept ($M-1$ keep-bits) the quantum is the value of the least-significant bit of keep-bits, i.e. at least $|2^{1-M}v|$, and at most $|2^{-M}v|$ depending on how far $v$ is from the maximum power of 2 that is less than $v$. Note that the quantisation levels for the *halfshave* method are in mid-points of the intervals between the levels for other methods.

The *shave* method implements rounding towards zero. The *set* method is almost rounding away from zero. They introduce an error of at most a full quantum to each value with a mean absolute error of a half-quantum. The *groom* method being alternately *shave* and *set* has the same absolute error as those, but is almost unbiased in average. The *round* method takes into account the value of the MSB of tail bits, an therefore introduces an error of at most a half-quantum with mean absolute error of a quarter of a quantum. Same applies to the *halfshave* method, since its levels are midpoints of the range controlled by the tail-bits of the original value. Note that for the mean-absolute error estimate we assume that the tail bits have an equal chance to be 0 or 1.

The margins of error introduced by the considered methods for different number of mantissa bits are summarized in Table. 1. As one can see, the *round* and *halfshave* methods allow for half the error for the same number of keep-bits or one less keep-bit to guarantee the same accuracy as *shave*, *set*, or *groom*.

The results of applying a precision-trimming with various number of keep-bits are illustrated in Fig. 1, where the 1:1 line is marked as "ref". The *set* method is not shown, to reduce number of lines in the panels. For the shown range and keep-bits of 8,

| mantissa size | keep-bits | discretisation level increment | | | max relative error | |
| --- | --- | --- | --- | --- | --- | --- |
| bit | | as a fraction of the value | | | *shave, set, groom* | *round, halfshave* |
| 1 | 0 | 0.5 | … | 1 | 100% | 50% |
| 2 | 1 | 0.25 | … | 0.5 | 50% | 25% |
| 3 | 2 | 0.125 | … | 0.25 | 25% | 12.5% |
| 4 | 3 | 0.0625 | … | 0.125 | 12.5% | 6.25% |
| 5 | 4 | 0.03125 | … | 0.0625 | 6.25% | 3.1% |
| 6 | 5 | 0.015625 | … | 0.03125 | 3.1% | 1.6% |
| 7 | 6 | 0.0078125 | … | 0.015625 | 1.6% | 0.8% |
| 8 | 7 | 0.00390625 | … | 0.0078125 | 0.8% | 0.5% |
| 9 | 8 | 0.001953125 | … | 0.00390625 | 0.5% | 0.25% |

**Table 1.** The discretisation level increment for different number of mantissa bits kept and maximum relative error introduced by trimming the precision with various methods

4, 6, and 2 the discrete levels are separated by 0.5, 2, 8, and 32 respectively. The levels of *halfshave* have half-discrete offset with respect to the others. Note the change of a discretisation-level spacing at the value of 256. For 8 keep-bits (Fig. 1a) on the scale of the plots the distortion is almost invisible, whereas for 2 keep-bits the difference is easily seen. Note the increase of the quantum size at 256 due to the increment of the exponent.

It was pointed by Zender (2016) the *shave* method introduces a bias towards zero, which might be unwanted in some applications (*set* introduces the opposite bias). The idea of Bit Grooming (*groom*) is to combine these two biased trimming procedures to get on average unbiased fields.

However, it has been overlooked that Bit Grooming introduces an oscillating component that affects multipoint statistics. With Bit Grooming, the quantization of a value in an array depends not only on the value itself, but also on its index in the array. As a result of that, the absolute difference between two values with even and odd indices will gets positively biased in average, while the absolute difference between two points with both even (or both odd) indices will stay unbiased.

One can note that the result of applying the *halfshave* procedure with the same number of keep-bits to a bit-groomed field is equivalent to the applying it to the original field. Therefore *halfshave* can be considered as a method to half the error and remove the artifacts of Bit Grooming.

## 4 Examples

Consider an array of $N$ floating-point numbers $u_i$ and its precision-trimmed version $v_i$. To illustrate the performance of the algorithms we will consider the normalised root-mean-square error (NRMSE) introduced by a precision-trimming

$$\text{NRMSE} = \sqrt{\frac{1}{N} \sum_{i=1}^{N} \frac{(u_i - v_i)^2}{u_i^2}}, \tag{2}$$

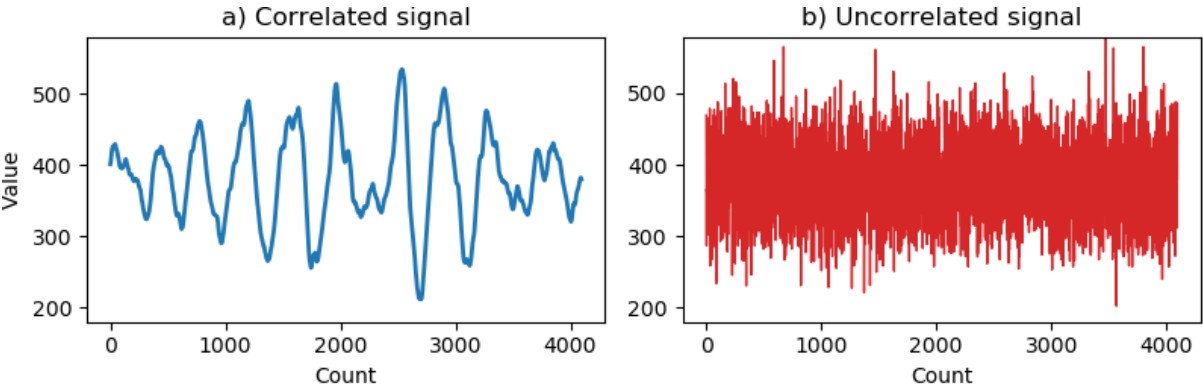

**Figure 2.** The signals used for illustration a) correlated and b) uncorrelated

|           | 2 keep-bit | 4 keep-bit | 6 keep-bit |
|-----------|------------|------------|------------|
| shave     | 0.09800    | 0.02479    | 0.00619    |
| groom     | 0.09941    | 0.02478    | 0.00620    |
| halfshave | 0.04968    | 0.01231    | 0.00310    |
| round     | 0.04927    | 0.01252    | 0.00311    |
| groomhalf | 0.04968    | 0.01231    | 0.00310    |
| groomav   | 0.04870    | 0.01141    | 0.00280    |

**Table 2.** NRMSE of the signal in Fig. 2a after trimming precision

and a distortion to the structure function of the precision-trimmed fields, which is defined as follows

$$X(r) = \sqrt{\frac{1}{N-r} \sum_{i=r}^{N-r} (u_i - v_{i+r})^2}, \tag{3}$$

where the integer argument $r$ is called offset, and can be spatial or temporal coordinate.

To illustrate the features of the precision-trimming we will use two synthetic arrays: a random process with "-2"-power
spectrum (Fig. 2a) and random Gaussian noise (Fig. 2b). The former is surrogate of a geophysical signal with high autocorrelation, whereas the latter corresponds to a signal with a large stochastic component. Both signals have identical variance, that is controlled cut-off at low-frequency components of the correlated signal, and by the variance parameter of the uncorrelated signal. To simplify the analysis we have added a background that is 8 times the signal's standard deviation. Each array is 32768 values long. The exact code used to generate the signals and all the figures of this paper can be found in the Supplementary
material.

The NRMSE of the signal in Fig. 2a after precision-trimming with the considered methods is summarised in Table 2. The results fully agree with Table 1: the NRMSE introduced by every method is about half of the maximum error for a single value.

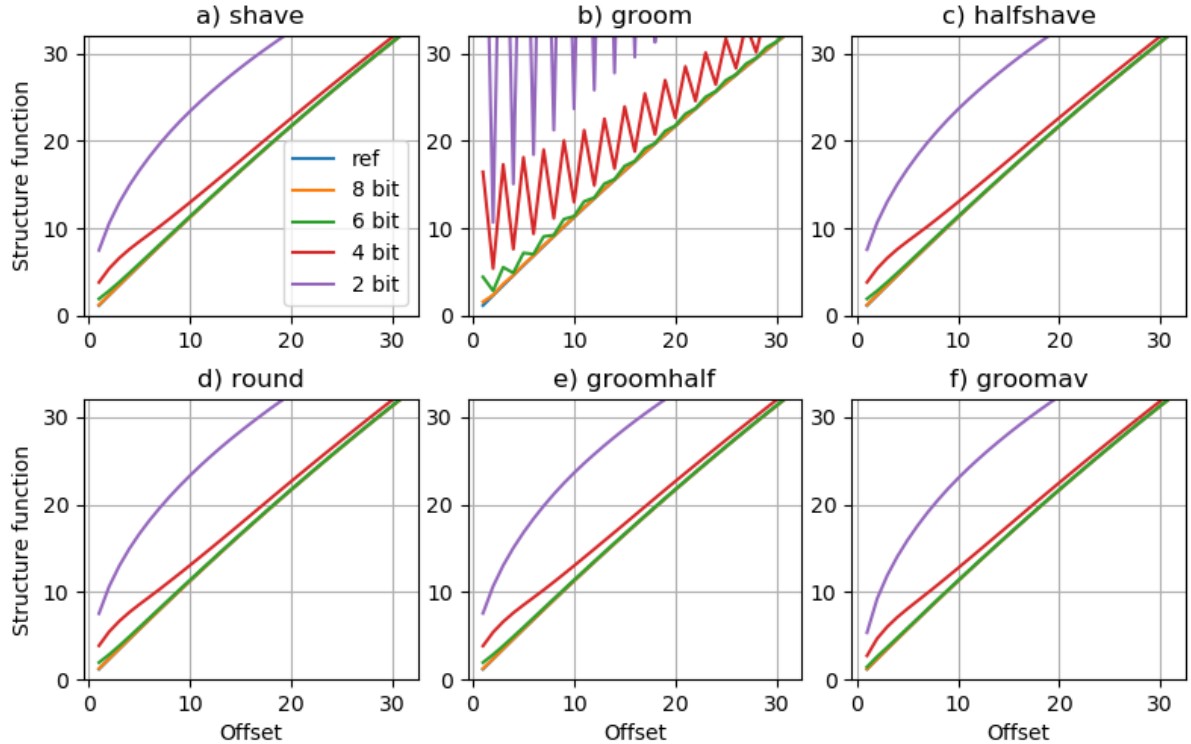

**Figure 3.** The structure functions (3) of the correlated signal shown in Fig. 2a, processed with different precision-trimmings

Along with above-mentioned methods we have added two ways of rectifying bit-groomed values: *groomhalf*, where we apply *halfshave* to the Bit-Groomed array, and *groomav*, where a simple two-point moving average is applied to the data.

As expected, *groomhalf* results in exactly the same values as *halfshave*, and therefore has identical NRMSE. The scores for *halfshave* and *round* are slightly different due to the limited sample size and limited number of discrete levels for the samples. It is notable that for smooth signals like one in Fig. 2a, the moving average of bit-groomed values gives smaller NRMSE than for all other methods. The reason is in the smoothness of the signal, which has well-correlated adjacent points, while bit-trimming errors are less correlated.

For the signal with −2-power spectrum (Fig. 2a) the structure function is linear. The structure functions of the signal processed with trim-precision are given in Fig. 3. All panels have the structure function of the original signal ("ref" curve) plotted along with curves for the processed signal. Since the structure function is not sensitive to the offsets, the plots for *shave*, *halfshave* and *groomhalf* (panels a, c, and e) are identical. Panel d differs slightly form them due to statistical differences mentioned above. The Bit Grooming algorithm produces quite large peaks at odd offsets, whereas the values at even offsets are equal to

the corresponding values for *shave* or *halfshave*.

As a way to compensate for the artifacts of bit grooming without introducing additional bias one could apply moving average (C. Zender, private communication). For a smooth signal running average produces a structure function that is closer to the

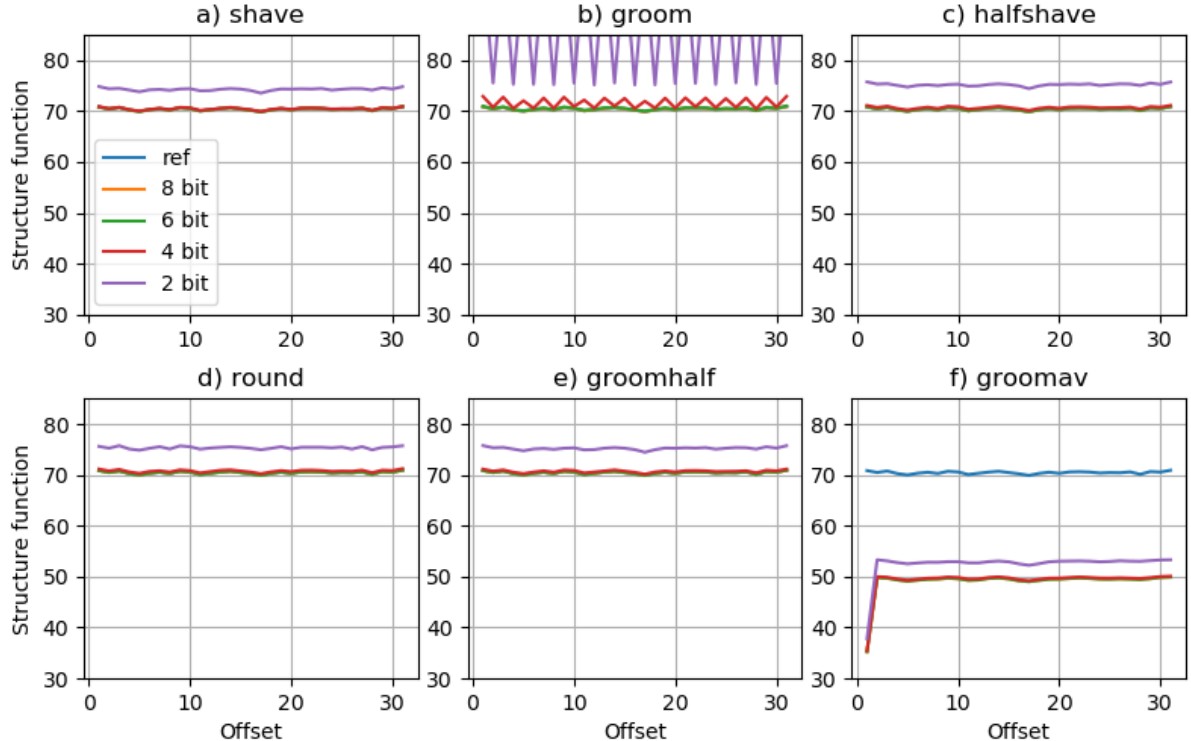

**Figure 4.** The structure function (3) of the signal shown in Fig. 2b, processed with different precision-trimmings

original one than the mantissa-rounding. The steeper structure function for smaller offsets in Fig 3f is a result of smoothing by the moving average.

The situation is different for the uncorrelated random signal from Fig. 2b, whose structure function is given in Fig. 4. For this signal the reference structure function is flat, and the remaining oscillations are due to the sampling errors. As in the previous case, the increase in the number of keep-bits makes the structure function closer to the reference one, but the offset also depends strongly on the location of the background and the variance with respect to the discretisation levels. As in the case of the smooth signal, Bit Grooming produces oscillating distortions of the structure function, which can be removed by applying *halfshave*.

The moving average notably reduces the variance of the uncorrelated signal, and the structure function gets lower. A dip at the unity offset at Fig 3f is caused by a correlation between neighbouring points introduced by the moving average. Therefore moving average for poorly-correlated signals is more damaging than Bit Grooming.

## 5 Keeping absolute precision

Some applications require a given *absolute* precision, i.e the acceptable error of data representation is expressed in absolute units rather than as a fraction of a value. For such data, when applying a rounding precision-trimming procedure, one can specify the value of the least-significant keep-bit, rather than the fixed number of mantissa bits, and therefore ensure that the absolute error of the representation will be within a half of that value. The latter method is called "decimal significant digit" (DSD-) method in NCO (Zender, 2016). Since both relative- and absolute- precision trimming methods reset least-significant bits of floating-point values, the statistical properties of the distortions introduced by these methods are similar.

If the required accuracy of a variable is given in terms of both relative and absolute precision, the precision-trimming with with limit for both least-significant keep-bit number and least-significant keep-bit value can be used. This can be achieved by sequential application of the two trimming procedures.

## 6 Precision of Linear Packing

Linear Packing (LP) is a procedure of applying a linear transformation that maps the range of the original values into the representable range of an integer data type, and rounding and storing the resulting integers. The parameters of the transformation, offset and scale factor, have to be stored for subsequent recovery of the data. The method itself leads to a reduction of the dataset size if the integer size is shorter than the original floating-point size. Applying a lossless compression to the resulting integer array usually leads to further reduction of size.

When applying LP to a dataset one has to select the parameters of the transformation and a size of the integer datatype (`bitsPerValue` parameter). This can be done in two ways:

1. One can specify `bitsPerValue` and map the range of an array into the representable range. In this case a full array has to be analysed prior to the packing, and the absolute precision of the representation will be controlled by the difference between maximum and minimum value of the array. This approach is acceptable when the encoded parameter is known to have a certain range of values, then the resulting errors can be reliably constrained.

2. One can explicitly specify the required absolute precision of the representation by pre-defining the scale factor. Then a number of `bitsPerValue` has to be taken with sufficient margin to ensure that all values can be mapped into the representable range.

In both cases LP controls the absolute error of the representation, which makes LP sub-optimal or for the data that require a constrain for a relative error. If a dataset has a large dynamic range (many orders of magnitude), the number of `bitsPerValue` might become larger than the bit-size of the original floating-point values, if at all representable with a specific output format (e.g. 32 bit for NetCDF).

GRIB format allows for applying LP to individual gridded 2D fields, and the `bitsPerValue` parameter can be selected to any value in in the range of $0 - 255$ (often only $0 - 63$ are implemented). Therefore LP parameters can be optimized for

each field to provide the needed absolute precision. Writing of large GRIB datasets can be done field-by-field in a single pass. The approach has been widely used and works well for meteorological data. Recent GRIB2 standards enable a lossless data compression of packed data with CCSDS/AEC (Consultative Committee for Space Data Systems / Adaptive Entropy Coding) algorithm, that provides further reduction of the packed data size for a typical output of a numerical-weather-prediction model by about a half (50–60% of reduction from our experience). This compression alleviates the use of redundant `bitsPerValue` when the absolute precision is specified (option 2 above), and therefore the GRIB2 implementation of LP is good for data of a limited dynamic range and a requirement for absolute precision.

LP for the classic NetCDF format has additional implications. The format provides a choice of 8, 16 or 32 bits per value for integers, and the integers are signed. According to the CF conventions http://cfconventions.org, LP applies the same linear transformation parameters for the whole variable in a file, that may contain many levels and/or time steps. Many implementations (e.g. `ncpdq` tool from NCO), use the option 1 from above, therefore require two passes through the data set, which might require consideration of memory constrains. This way usually does not lead to noticeable artifacts for common meteorological variables. However, if the original field has large dynamic range or has substantially skewed distribution the errors of LP can be huge.

Consider a gridded field of in-air concentration of a pollutant continuously emitted from a point source in the atmosphere. The maximum value of such a field would depend on a grid resolution, and he number of bits needed to pack such a field with a given relative or absolute accuracy has to be decided on a case-by-case basis. Moreover, if one would need to integrate such a field to get total emitted mass of the pollutant, the integral of the unpacked field can be very different from the integral of original field. If the zero value of the original field can be exactly represented as packed value, a substantial fraction of mass can be hidden in near-zero values. If the zero value of the original field is mapped to a ("small") non-zero value the total mass, obtained from integration of the unpacked field would depend on the domain size.

Rounding errors in linear packing can lead to unphysical values. Consider a single-precision field (of e.g. humidity) that has values between 0 and 0.99999535. If one packs it to a 8-bit signed integer (byte) with standard procedure of NCO (as in version 4.9.4) and unpacks it back with the same NCO, one gets $-2.980232 \cdot 10^{-8}$ as the minimum value instead of zero. The negative value results from rounding errors and a finite precision of the packing parameters. An unphysical value used in a model might lead to arbitrary results. If the variable has a specified `valid_range` attribute, CF conventions prescribe to treat corresponding values as missing.

Therefore we conclude that LP produces acceptable results when the required margin for absolute errors is within a few orders from the field's maximum value, and if the valid range of a variable covers the actual range with that margin. In general case the application of LP might lead to errors of arbitrary magnitude, or can turn valid data invalid.

A useful application of the packing parameters is scaling or unit-conversion for the data that already have trimmed precision. If one multiplies precision-trimmed data with a factor that differs form a power of two, the scaling affects also the tail-bits, and the compression efficiency reduces. However, if one scales the data by altering the `scale_factor` attribute of a NetCDF file the stored array stays intact and well compressible, and CF-aware readers will apply the scaling on reading.

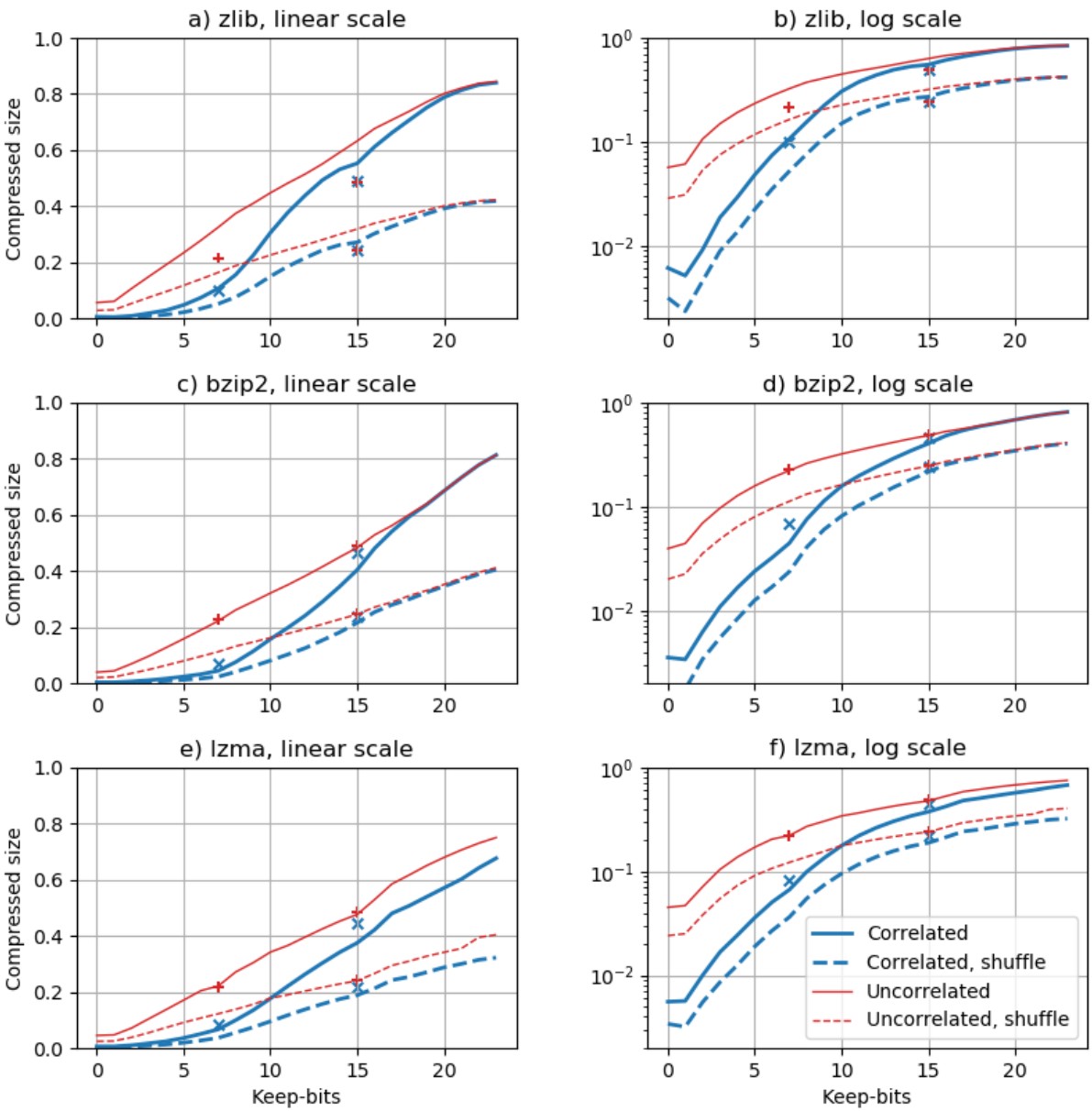

**Figure 5.** The normalised compressed size (compression ratios) of the arrays shown in Fig. 2, as a function of the number of keep-bits for mantissa-rounding for two test arrays with and without shuffling. Compression ratios for same arrays packed with Linear Packing are shown with points of respective color. The ratios are shown both in linear and logarithmic scale for three compression methods. 23 keep-bits correspond to the original precision.

## 7 Compressing precision-trimmed data

To give an example of the space gain due to compression we compressed the test arrays shown in Fig. 2 processed with mantissa rounding, and plotted the compressed-data size as a function of the number of trimmed bits. The size is normalized with the storage size of the original array ($32768 \cdot 4$ bytes). We used three compression methods available from the standard Python library: *zlib* (the same as used with NetCDF4 compression) with level 6, *bzip2*, and *lzma*. The latter two are more expensive computationally but provide higher compression ratio, and can be applied to compress the raw-binary data files, or used in

future data formats.

To emulate the compression used with NetCDF4 we have applied *shuffling* to the bit-trimmed arrays. Shuffling is a procedure of re-arrangement of the data bytes in a floating-point array, so they get ordered sequentially: all first bytes of each number, then all second bytes etc. The procedure is known to improve the compression ratio in most cases. Shuffling has been available in NetCDF4 library and is applied in NCO in writing compressed variables.

Fig. 5 shows the compression ratio for the precision-trimmed test signals with (dashed lines) and without (solid lines) shuffling processed with three compression methods. We used mantissa rounding with number of keep-bits form 0 to 23, i.e. full range of possible trimmings for single-precision floats. To better visualize the full range of compression ratios we plot them in both linear and logarithmic scales. For a reference the compressed-size of the same arrays packed with linear packing to 8- and 16-bit signed integers is shown. The packed data are attributed to the number of keep-bits corresponding to the same

maximum absolute error.

For all algorithms the reduction of keep-bits leads to the reduction of compressed size. The reduction is nearly linear for the uncorrelated signal and faster than linear for the correlated signal, since more significant bits of the latter are less likely to differ between neighbouring points. For the example arrays the monotonicity breaks between 0 and 1 keep-bits since for 0 keep-bits the rounding procedure alters the mantissa, which stays otherwise constant. Some of the curves have a noticeable dip

in the compressed size corresponding to 7 and 15 keep-bits, corresponding to byte-boundaries. The dip is mostly pronounced for *zlib* compression.

Shuffling substantially improves the compression efficiency, and leads to twice smaller resulting size for our datasets for all compression methods, including Linear Packing to 16-bits. The reason is that general-purpose compression algorithms operate on the data as a sequence of bytes, and grouping of corresponding bytes facilitates the recognition of repeating patterns in the

285 data flow.

At high precisions the compression algorithms do not take advantage of the correlated signal and both signals have the same compression ratio, except for *lzma*, which is the most expensive of all three. The difference between the signals is also small for 16-bit Linear Packing. At lower precisions the correlated array compresses notably better than the uncorrelated one.

Compression of Linearly-Packed data does only marginally better for the correlated array at 16-bit, while makes a notable

difference between the arrays at 8-bit resolution. Linear Packing of the uncorrelated array at 16-bit leads to some 15% better *zlib* compression than corresponding precision-trimming, for the correlated array the difference is much smaller. In all other cases the precision trimming results in the same or better compressions than corresponding shuffled packed array.

## 8   Practical examples

Lossy compression is a compromise between the accuracy and the compression efficiency. Optimal parameters of compression depend on the nature of the data and on the intended application. In this section we give examples on suitable compression parameters for several practical applications. For illustrations we use some atmospheric variables, that the author dealt with.

Consider gridded fields of the 2-meter temperature and the surface-air pressure in the Earth atmosphere. The fields have a limited dynamic range: the values of pressure fall within 330 hPa (at the Everest summit) and 1065 hPa (at Dead Sea level), and for the temperature the range is 200 to 360 K. For such fields the spacing of quantization levels for precision-trimming falls within a factor of 2-4, and to achieve the same norm of absolute (or relative) precision one would need approximately the same number of mantissa bits for precision trimming or bits-per-value for Linear Packing. The fields are usually smooth, so the most-significant bits do not differ among neighbouring cells, and a subsequent lossless compression yields a substantial gain in size even with large number of bits kept.

Suppose, the intended use of the dataset is calculation of the components of a geostrophic wind $u_g$, $v_g$, defined as (Holton and Hakim, 2013):

$$-fv_g \simeq -\frac{1}{\rho}\frac{\partial p}{\partial x}; \qquad -fu_g \simeq -\frac{1}{\rho}\frac{\partial p}{\partial y}, \tag{4}$$

where $f \simeq 10^{-4}$ s is a Coriolis parameter, and $\rho$ is the air density. If the target accuracy for the geostrophic wind is 0.1 m/s or 1%, and the grid spacing is 10 km, one would need the pressure accurate within 0.01 hPa, and temperature within 2 K. Therefore 17 mantissa bits will be needed for the pressure, while only 6 bits are needed for the temperature. However, if both fields are needed to convert between the volume mixing ratio and the mass-concentration of an atmospheric trace gas, whose measurements are accurate within e.g. 1%, 6 to 7 bits per value will be sufficient for both variables.

Another example is wind components from a meteorological model. For practical applications, like civil engineering or wind energy one would need a precision that is better than one of reference wind measurements. Best atmospheric anemometers have about 1% relative accuracy and about 1 cm/s absolute accuracy. If we neglect representativeness error, which is usually larger than an instrument error, but less trivial to evaluate, one would need 6 to 7 mantissa bits for the model wind components. If one is going to reconstruct the vertical wind component from the horizontal ones using the continuity equation, e.g. for offline dispersion modelling, the needed precision of horizontal components is absolute and depends on the grid resolution, similarly to the aforementioned case of geostrophic wind.

One could note, that relative precision-trimming leaves excess precision for small absolute values of wind, and therefore is suboptimal for data compression. The absolute-precision trimming should be more appropriate for wind components. Since the range of wind components in the atmosphere is limited, linear packing has been successfully used for wind components without major issues. Archives of NWP forecasts typically use Linear Packing with 16-18 bits per value for wind components.

In the aforementioned case of a concentration field originating from a point source, where LP is inapplicable, precision-trimming works well. The precision-trimming does not affect representable dynamic range of the field, and relative the precision of the integral of the field over some area is at least the same as one of the precision-trimmed data.

Often, in problems of atmospheric dispersion, both the emission sources and observations have quite large uncertainty, therefore there is no reason to keep large number of significant figures in simulation results. If one creates a lookup database for scenarios of an emergency release of hazardous materials at various meteorological conditions, the database size is one of the major concerns, while the uncertainty of the released amount is huge. For such applications even 1-2 keep-bits for the concentration fields can be sufficient.

For model-measurement comparisons of such simulations one might be interested only in concentrations exceeding some threshold, e.g. some fraction of detection limit of the best imaginable observation. Then in addition to trimming relative precision, one could trim the absolute precision of the field and therefore zero-out substantial part of the model domain, further reducing the dataset storage size.

## 9   Conclusions

A simple method for trimming precision by rounding a mantissa of floating-point numbers has been implemented and tested. It has been incorporated into NCO mainstream and is used by default since v4.9.4. The method has half the quantization error of the Bit Grooming method (Zender, 2016), which was used by default in earlier versions of NCO. Bit Grooming, besides having suboptimal precision, leads to substantial distortion of multipoint statistics in scientific data sets. The "halfshave" procedure can be used to partially recover the precision and remove excess distortions from two-point statistics Bit-Groomed datasets.

Precision trimming should be applied to data arrays before feeding them to NetCDF or any another data-output library. The trimming can be applied to any data format, e.g. raw binary, that stores arrays of floating-point numbers to facilitate subsequent compression. NCO provides a limited leverage to control the number of mantissa bits in terms of specifying "a number of significant digits" (SD) for a variable. These digits can be loosely translated into the number of mantissa bits: 1 SD is 6 bits, 2 SD is 9 bit, 3 SD is 12 bits etc. The exact translation slightly varies among the NCO versions, therefore a low-level data processing should be used if one needs to control exact number of mantissa bits to keep.

Along with a relative-precision trimming that keeps a given number of mantissa bits, and therefore guarantees a given *relative* precision, we considered absolute-precision trimming that specifies a value for the least-significant mantissa bit. The latter method is recommended when required *absolute* precision is specified. Depending on the nature of a variable in the dataset and intended application ether or both of the trimmings can be applied to remove non-significant information and achieve the best compression performance.

Precision-trimming and subsequent lossless compression has substantial advantages over the widely-used Linear Packing method: it allows to explicitly specify the needed precision in terms of both absolute and relative precision, guarantees to keep the sign and a valid range of an initial value and allows for use of the full range of floating-point values. Our examples illustrate that Linear Packing can lead to unconstrained errors and does not provide substantial savings in the storage space over precision-trimming, therefore Linear Packing should be considered deprecated. The exception is GRIB2, where Linear Packing is applied to individual 2D-fields of known dynamic range, uses unsigned integers of arbitrary length, and involves specialized compression algorithms.

The precision-trimming methods described in the paper were implemented in Python, and corresponding subroutines are given in the Appendix, where we also put subroutines for relative- and absolute-precision trimming with mantissa rounding, implemented with Fortran 95 intrinsics. The subroutines should be directly usable in various geoscientific models. The implementations can be feely used under the BSD license.

*Code availability.* The implementation of mantissa rounding in C has been merged to the NCO master branch. The source code of NCO is available from http://nco.sourceforge.net (last accessed December 7, 2020) under BSD License. The Python3 code used to generate the figures and the example statistics is available from the supplementing material and can be used under the BSD License.

## Appendix A: Implementation of precision-trimming

In the appendix we list simple subroutines that implement *shave*, *half-shave* and two version of *round* precision-trimming. The subroutines can be directly copy-pasted or easily adapted for practical applications. As it is pointed in Conclusions, round half to even should be used for precision trimming unless there is a good reason to apply another method. The subroutines are implemented in Python/numpy and can be embedded directly to a data-handling software. Also rounding implemented in Fortran 95 without explicit bitwise operations is given below.

All subroutines receive an array a of single-precision floats to process and an integer `keepbits` specifying the needed number of keep-bits for the output array. The subroutines can be adapted to handle double-precision floats as well.

We assume that all the parameters are within a valid range: the number of keep-bits is within the range from 1 to 24, and the array values are finite and not equal to any value with a special meaning in the context of the application (e.g. missing value).

### A1 Bit Shave

Here a simple mask zeroing tail-bits is applied to the array.

```
import numpy as np
def shaveArray(a, keepbits):
    assert (a.dtype == np.float32)
    b = a.view(dtype=np.int32)
    maskbits = 23 - keepbits
    mask = (0xFFFFFFFF >> maskbits)<<maskbits
    b &= mask
```

### A2 Half-Shave

Same as above, but the most-significant of tail-bits is set.

```
import numpy as np
```

```
      def halfShaveArray(a, keepbits):
          assert (a.dtype == np.float32)
b = a.view(dtype=np.int32)
          maskbits = 23 – keepbits
          mask = (0xFFFFFFFF >> maskbits)<<maskbits
          b &= mask
          b |= (1<<(maskbits-1))
```

## A3   Rounding, half-to-infinity

Here we use an integer operation to add a half-quantum before bit-shaving.

```
      import numpy as np
      def TrimPrecision(a, keepbits):
          assert (a.dtype == np.float32)
b = a.view(dtype=np.int32)
          maskbits = 23 – keepbits
          mask = (0xFFFFFFFF >> maskbits)<<maskbits
          half_quantum = 1 << (maskbits – 1)
          b +=  half_quantum
b &= mask
```

## A4   Rounding, half-to-even

To implement half-to-even, before bit-shaving we add half-quantum to those values that have 1 in the least significant of keep-bits, and they get rounded half-to-infinity as in the previous example. If the least significant of keep-bits is 0, we add less-than half-quantum, so bit-shaving results in rounding half-to-zero. This code was inspired by Milan Klöwer (private communication).

```
      import numpy as np
      def TrimPrecision(a, keepbits):
          assert (a.dtype == np.float32)
          b = out.view(dtype=np.int32)
maskbits = 23 – keepbits
          mask = (0xFFFFFFFF >> maskbits)<<maskbits
          half_quantum1 = ( 1 << (maskbits – 1) ) – 1
          b +=  ((b >> maskbits) & 1) + half_quantum1
          b &= mask
```

## A5 Rounding, half-to-even in Fortran 95

Since type-casting is not reliable in some versions of Fortran, here is an implementation that uses only standard Fortran 95 intrinsics. It is probably not as efficient computationally as an implementation via bitwise/integer operations, but the cost of this operation should be a negligible in a context of any geophysical model. The subroutine uses default rounding mode of Fortran, which is half-to-even, according to IEEE 754.

```
subroutine trim_precision(a, keepbits)
      real (kind=4), dimension(:), intent(inout) :: x
      integer, intent(in) :: keepbits
      real :: factor
```

```
      factor = 2 ** (keepbits + 1)
      a =  nint(factor * fraction(a))*(2.**exponent(a))/factor
```

```
end subroutine trim_precision
```

## A6 Rounding to given absolute precision in Fortran 95

In this example `max_abs_err` specifies the limit for absolute error for each individual value. The resulting absolute error is within a half of `quantum`, which is set to the integer power of two to ensure that tail-bits are set to zero. As in the previous example the half-to-even IEEE 754 rounding produces the unbiased result.

```
subroutine trim_absolute_precision(a, max_abs_err)
      real (kind=4), dimension(:), intent(inout) :: x
real, intent(in) :: max_abs_err
      real :: quantum, factor, a_maxtrimmed
```

```
      quantum = 2**(floor(log2(max_abs_err))+1)
      a_maxtrimmed = quantum * 2**24 !! Avoid integer overflow
```

```
      where (abs(a) < a_maxtrimmed) a =  quantum * nint(a / quantum)
```

```
end subroutine trim_precision
```

*Competing interests.* The author declares no competing interests

*Acknowledgements.* I would like to thank Prof. Mikhail Sofiev, and Dr. Viktoria Sofieva from the Finnish Meteorological Institute, and Dmitrii Kouznetsov from the University of Electro-Communications, Tokyo for fruitful discussions, and Dr. Charles Zender from the University of California for the NetCDF operators software (NCO) and for being open and quick for verifying and accepting the patches for it. Also I am grateful to Dr. Milan Klöwer from the Oxford University, Dr. Mario Acosta from the Barcelona Supercomputer Center, Dr. Seth McGinnis from the National Center for Atmospheric Research, and Dr. Ananda Kumar Das from the Indian Meteorological Department for

their careful reading of the discussion paper and their critical and valuable comments.

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
