# Peer review of "A note on precision-preserving compression of scientific data"

_Geoscientific Model Development, 2020_

## Short Comment (SC1) · 30 Jul 2020

The author discusses shortcomings of the some rounding methods for floating-point numbers used in data compression techniques implemented in the NCO software library. The points adressed are worthwhile as the default rounding mode in NCO, so-called BitGrooming, introduces artifacts that can be avoided with other techniques, as shown in this manuscript. The author proposes a round-to-nearest mode as a default rounding mode and provides evidence for its advantages over the other rounding modes such as BitGrooming, -Shaving, and halfshave. Methods to remove the shortcomings of bitgrooming from a previously groomed dataset are presented additionally.

[Figure]

**1 General remarks**

**1.1 Tie rules**

The author introduces a round-to-nearest mode that is based on 1 floating-point addition, 1 multiplication and two shave operations, which will be called the 2u-v method from now on, as follows from the underlying equation in the manuscript. The manuscript is currently lacking a discussion on the tie rules, i.e. the behaviour of the 2u-v rounding when a float is exactly between two representable quantums. If the distance between those is 1 ULP than I am referring to floats at 1/2 ULP. The 2u-v method is equivalent to round-to-nearest tie away from zero, which is bias-free for data that is symmetrically distributed around 0, but introduces a bias for data that is predominantly positive or negative. I therefore propose to include this information in the manuscript to distinguish the 2u-v method from the IEEE round-to-nearest tie-to-even (aka half to even) standard. The author can make a point though, that in the case of data compression rounding errors usually don't accumulate as they can in e.g. simulations of dynamical systems, and therefore might be negligible. In any case, it is proposed to include a discussion around this bias (where it is also important to clarify that a tie-bias is different to the bias introduced by bit-shaving for example, which scales with ULP) and whether it is important in the typical use-cases of NCO.

I suggest to discuss how your method is different from round-to-nearest tie-to-even which is already mentioned in IEEE-754 from 1985 as "An implementation of this standard shall provide round to nearest as the default rounding mode. In this mode therepresentable value nearest to the infinitely precise result shall be delivered; if the two nearest representable values areequally near, the one with its least significant bit zero [which is the "even" number] shall be delivered." (page 5 therein)

To illustrate this remark, consider the following float32 numbers. round refers here to round-to-nearest tie-to-even ("half to even") and kouzround to the 2u-v method intro-

duced by the author.

```
julia> a
5-element Array{Float32,1}:
 0.09765625
 0.0041503906
 0.0061035156
 0.028320312
 0.033203125
```

In its bit-representation they are (split into sign,exponent and significant bits)

```
julia> bitstring.(a,:split)
5-element Array{String,1}:
 "0 01111011 10010000000000000000000"
 "0 01110111 00010000000000000000000"
 "0 01110111 10010000000000000000000"
 "0 01111001 11010000000000000000000"
 "0 01111010 00010000000000000000000"
```

Rounding these numbers with round-to-nearest tie-to-even (round towards zero in this case) and keeping 3 significant bits yields

```
julia> bitstring.(round(a,3),:split)
5-element Array{String,1}:
 "0 01111011 10000000000000000000000"
 "0 01110111 00000000000000000000000"
 "0 01110111 10000000000000000000000"
 "0 01111001 11000000000000000000000"
 "0 01111010 00000000000000000000000"
```

In contrast, the 2u-v method (aka "kouzround") rounds these ties away from zero

```
julia> bitstring.(kouzround(a,3),:split)
5-element Array{String,1}:
 "0 01111011 10100000000000000000000"
 "0 01110111 00100000000000000000000"
 "0 01110111 10100000000000000000000"
 "0 01111001 11100000000000000000000"
 "0 01111010 00100000000000000000000"
```

In the case where the even quantum is away from zero, both methods are indeed identical

```
julia> b
0.030273438f0

julia> bitstring(b,:split)
"0 01111001 11110000000000000000000"

julia> bitstring(round(b,3),:split)
"0 01111010 00000000000000000000000"

julia> bitstring(kouzround(b,3),:split)
"0 01111010 00000000000000000000000"
```

Note how the carry bit carries correctly into the exponent bits, something that bitgrooming, shaving and setting would never do.

**GMDD**

**1.2 Implementing round-to-nearest tie-to-even with bitwise operations is "difficult"**

Although I agree that the 2u-v method is fairly simple to understand conceptually and therefore intuitively easier to implement, a bitwise round-to-nearest (tie-to-even) still only requires 1 shift, 2 bitwise-AND, and 2 integer-ADD. I would therefore argue that its implementation is of even lower algorithmic complexity than 2u-v rounding, which involves, despite hidden, way more bitwise operations within calls to float-multiply and float-subtract. Assume you want to keep 7 significant bits of a float32 number, intepreted here as a unsigned integer ui, meaning there are 16 tailing bits. The algorithm then reads

```
ntb = 16                                 # number of tailing bits, 16 here
shavemask = 0x0000_ffff      # cover all tailing bits
mask2 = 0x0000_7fff            # cover all tailing bits except the most signifcant

**account for the carry bit**
ui += mask2 + ((ui >> ntb) & 0x0000_0001)
ui &= shavemask                          # shave tailing bits
```

Which can be easily generalised for various numbers of keepbits. It is still very much worthwhile to introduce your method, such that libraries similar to NCO can make use of it. However, if you introduce a rounding method that does not

**2 Minor remarks**

(i) L.48-49

*the least-significant bits (LSBs) of mantissa contain arbitrary, often chaotic information,which makes lossless-compression algorithms inefficient"*

in comparison to L.19-20

*non-significant bits of the value scontain arbitrary numbers with high entropy, that are difficult to compress*

I agree with you that the tailing bits contain seemingly random or arbitrary bits, which have a high entropy and are therefore difficult to compress. I use the words "seemingly random" as these bits don't have to be truly random but at least somewhat irregular. I find it problematic to use "chaotic" as also non-chaotic systems can produce those. E.g. even the exp-function will produce

```
julia> bitstring.(exp.(Float32.(1:10)))
10-element Array{String,1}:
 "01000000001011011111100001010100"
 "01000000111011000111001100100110"
 "01000001101000001010111100101110"
 "01000010010110100110010010000010"
 "01000011000101000110100111000101"
 "01000011110010011011011011100011"
 "01000100100010010001010001000011"
 "01000101001110100100111101010100"
 "01000101111110100011100010101100"
 "01000110101011000001010011101110"
```

As $e$ is transcendental, so is $e^2$ etc. which also means that in binary they will have an irregular sequence of bits (at least in the significant). I therefore suggest to change your wording slightly to give the reader a better understanding of what "difficult to compress" means: (i) Seemingly random, such that a loss-less algorithm can't identify a pattern and (ii) high entropy, such that entropy-encodings will not yield any benefits.

---

## Author Comment (AC1) · 15 Aug 2020

First of all I would like to thank you for constructive comments.

Indeed the implemented rounding introduces a bias of 1/2 ULP of the initial value, i.e about $10^{-7}$ for a single-precision float, and about $10^{-16}$ for double-precision. That fact is worth a note, but I doubt if more advanced rounding techniques are needed in precision-trimming, since, as it has been pointed, the errors of precision-trimming do not accumulate. I am not sure if three extra operations, needed to implement tie-to-even, as suggested in your comment are worth getting rid of an insignificant bias. If one really cares about $10^{-7}$ bias in single-precision, why would they use single precision, and why to apply precision trimming at all?

[Figure]

2u-v method was the first trial, which is quite simple to understand. The rounding method currently implemented in nco (4.9.4-alpha07) uses integer arithmetics, namely adding a half-shave mask as an unsigned integer and then applying a bit-shave mask. As it has been pointed in the comment, when adding the mask as unsigned-integer operation, the carry bit leads to the right result even if it reaches the exponent.

Thank you for suggesting the term "seemingly random", that seems to be the right one. I will use that.

As a side note, the comment states that rounding has advantages over halfshave. Indeed that has not been shown. The error norm and computational costs and a bias of halfshave are identical to those of rounding (as it has been implemented, i.e. tie-away-from-zero). The only advantage of rounding is of mostly aesthetic nature: it keeps small integer values intact. Probably, this has to be specified more clearly in the paper.

Also "shavemask = 0x0000_ffff" in the code example from the comment should read as "shavemask = 0xffff_0000".

---

## Referee Comment (RC1) · Mario Acosta (Referee) · 7 Sep 2020

This paper explains the limitations for data compression of one default precision-trimming method used for NetCDF and propose a simple but effective way to improve it using a mantissa-rounding technique, a novel idea which improves significantly the precision results. Moreover, the paper proves that this Bit Grooming algorithm has suboptimal accuracy and produces substantial artifacts in multipoint statistics. Additionally, they suggest a way to rectify the data already processed with Bit Grooming.

The new technique presented should be interesting for the community and novel enough for publication, being one of the default methods used by NetCDF. I would recommend extending significantly the state of the art and some rewriting of the sections for a better comprehension before this paper could be published and to support the novelty contribution.

One main question that the author would explain here before publication. How should the rounding bit size would be decided? Is this a parameter so set up through NetCDF?

Will the precision depend on the particular application or not? Apart from the array examples presented, the author should explain more in detail this.

Please find other requirements according to the section of the paper: line 14: Consider 32 bits as float can be confused since this depends on the program language. Actually, some languages define float as double precision, using integer (int) for this declaration. If a more in detail explanation is not done, I would remove float and double. lines 17-23: Have you consider different scientific fields to affirm that level of accuracy (less than 7 bits) or entropy? I agree that in most cases single precision is more than enough but you should enumerate cases where this could not be true and provide a state of the art about this. As an example, I consider that in the inputs for data assimilation of chaotic applications, as weather models, the level of accuracy or entropy to ensure the number of bits used could affect the results. Introduction: I consider that the state of the art should be extended. In order to support the argument explained in the second paragraph, some successful examples should prove that the reduction of precision does not affect to the accuracy. Both of them should be listed, applications using the reduction of precision and the reduction of precision to save data. Introduction: Is precision-trimming the single compression method available? Apart from a default method, is it the most used or one of the best methods for Netcdf files? The author should explain why this particular method and why the comparison done should be interesting for the reader, including some state of the art about this (apart from the single reference provided). The mantissa-rounding technique explained here should highlight if it is novel or there are other works which present similar approaches, or at least, differences among them. Section 2 should be rewritten. Second paragraph is difficult to follow and the explanation about the problem of trimming LSBs difficult to

understand. Five methods are described but only three of them are commented and only two discussed in details without more explanations about the reasons. It is also not clear how these results in Figure 1 are obtained. Moreover, it is not clear if these results and the explanation given is coming from Zender (2016) paper or from Section 3 and 4 of this paper. The author should highlight the novel contribution of this paper for the different sections.

As minor details about the figures, X and Y axes for all of them do not contain units. Figure 2: a) and b) are difficult to see.

---

## Referee Comment (RC2) · Seth McGinnis (Referee) · 11 Sep 2020

**GENERAL COMMENTS**

This paper describes some undesirable effects of the lossy compression algorithms currently implemented in NCO, and presents a correction that avoids these problems. This paper is well-structured, clearly written, logically sound, and of interest to the GMD community. The scientific significance, quality, and reproducibility are all very good; the only recommendations I have to make are minor revisions regarding presentation quality. Good work!

**SPECIFIC COMMENTS**

1) The subject is a little dry; I think the paper would benefit from a brief discussion in

the Introduction of the real-world use-case that led to the discovery of the two-point distortion, which would help the reader to follow the analysis using the structure function and understand why the issue matters.

2) The figure placement is a bit off. The figures should come shortly after their first mention in the text. This could be a problem that will be solved when the article goes to press, in which case please disregard this comment.

3) Figures 3 and 4 need axis labels. I would also recommend placing the legend outside the panels.

4) Also in Figures 3 and 4, what does "bg=300" mean in the title of each panel? If it's not necessary, it should be omitted; if it is needed, put it only in one place, perhaps in the caption.

5) In the captions for Figures 3 and 4, mention that the structure function is defined in section 4 of the text.

TECHNICAL CORRECTIONS

**Line: change from this -> to this**

2: magnitude keeping -> magnitude while keeping

5-6: use of a lossy compression -> use of lossy compression

8: twice higher precision -> double the precision

17: only few (or even few tens -> only a few (or even a few tens

18: As a result application -> As a result, application

19: dataset size since -> dataset size, since

20: high entropy, that -> high entropy, which

22,27,28,37,38,94,118: precision trimming -> precision-trimming

26: 2020) work efficiently -> 2020) to work efficiently

28: It has been quickly noticed hat -> It was quickly noticed that

28: implementation of a precision -> implementation of precision

33: algorithms and inaccuracies -> algorithms and the inaccuracies

33: Our analysis revealed -> Our analysis reveals

34: distortion two-point structure -> distortion to the two-point structure

34: fields, caused -> fields caused

35: twice higher accuracy -> double the accuracy

36: Besides that, -> In addition,

37: a method ... is suggested -> we suggest a method

41: extrapolated on -> extrapolated to

51: numbers set transform it so, that -> numbers, transform it so that

53: for N most-significant -> for the N most-significant

53: mantissa, that we let to store -> mantissa, which we use to store

54: "tail bits" the remaining -> "tail bits" for the remaining

60: ones -> one

63: rounding of mantissa -> round mantissa

64: have been implemented -> were implemented

64: and described well -> and are described well

66: floating-pint -> floating-point

76: kept in mantissa -> kept in the mantissa

85: methods of above -> methods above

85: correspondingly -> respectively

86: full quantum -> a full quantum

86: with mean absolute -> with a mean absolute

87: alternate -> alternately

88: at most half-quantum -> at most a half-quantum

89: Same applies to the he -> The same applies to the

90: have equal chance -> have an equal chance

91: margins for an error -> margins of error

93: method -> methods

93: twice smaller error -> half the error

96: correspondingly -> respectively

97: respect to others -> respect to the others

97: in the scale -> on the scale

98: keep bits -> keep-bits

98: well seen -> easily seen

104: get a positive bias in average -> be positively biased on average

106: to compensate -> to compensate for

107: This procedure, however -> However, this procedure

109: a way to half -> a method to halve

110: and to remove -> and remove
113: consider normalized -> consider the normalized

119: and a random -> and random

120: whereas, the latter -> whereas the latter

120: with high stochastic -> with a large stochastic

120: array is of -> array is

123: summary of NRMSE -> summary of the NRMSE

123: trimming precision -> precision-trimming

124: Table. 1: -> Table 1:

126: average applies -> average is applied

130: Notable, that -> It is notable that

131: signal that has -> signal, which has

135: sensible -> sensitive

136: slightly differs form -> differs slightly from

137: equal to corresponding -> equal to the corresponding

138: the structure functions -> a structure function

141: increase of the number -> increase in the number

142: reference one, however -> reference, but

144: that can be -> which can be

145: due to its the -> due to its

148: twice smaller -> half the

148: error than a -> error of the

148: 2016), that has -> 2016) which has

151: two-point statistics -> two-point statistics in

---

## Referee Comment (RC3) · Ananda Kumar Das (Referee) · 14 Sep 2020

Major Comments: The paper described a method which is simple but its utility has great implication in practical and real problem of data archival. A little bit of description may be added in the introduction to establish the need of such a method. As the same has been mentioned by other reviewers, the elaboration on this point is skipped here. The methodology part of the paper which describes different precision algorithm based on the static specification of the bits of mantissa is lacking a little visibility without schematics through diagram. The diagram showing distribution of values in the 32 bits may be schematically presented for a broad class of readers. If within the scope of the paper, the use of the specified method within NCO for real data may be tested and test result statistics may be added in the appendix.

[Figure]

Minor Comments: The label of the Fig 2 (b) is not visible as it is hidden within the plots. The same may be rectified.

---

## Author Comment (AC2) · 1 Oct 2020

Thank you very much for for constructive comments!

The main question of the review is: "How should the rounding bit size would be decided? Is this a parameter so set up through NetCDF?"

There seems to be a confusion between NetCDF (data format), libnetcdf (library to read/write it), and NCO (a popular software to manipulate the data in NetCDF format). Decision on the number of necessary mantissa bits is non-trivial. It depends strongly on the nature of the data and on the application, but has little relevance to the specific data format (NetCDF). The number of mantissa bits should not be decided by any general-purpose data handling software or library, but rather should be carefully selected by an

experienced user or by the author of a specialized software.

Precision trimming is an operation that should be applied to data arrays before feeding them to NetCDF or another data-output library. It can be applied to any data format, e.g. raw binary, that stores arrays of floating-point numbers to facilitate subsequent compression. CDO provides a limited leverage to control the number of mantissa bits in terms of specifying "a number of significant digits" (SD) for a variable. These digits can be loosely translated into the number of mantissa bits: 1 SD is 6 bits, 2 SD is 9 bit, 3 SD is 12 bits etc. The exact translation slightly varies among the CDO versions.

The above considerations will be incorporated in the revised version of the paper. The specific points will be addressed there as well, and summarized in the final response.

---

## Author Comment (AC3) · 1 Oct 2020

Thank you very much for your positive response and suggestions on improving the narrative and fixing language/technical issues! Very appreciated!

The "dryness" of the subject has been caused by my intent to focus on general properties of the precision-preserving compression without distracting attention to my specific use cases. However adding a couple of sentences on the use-case without very specific details would do no harm, so they will be added.

Other your points will be addressed in the final response and the revised paper.

[Figure]

2020.

---

## Author Comment (AC4) · 1 Oct 2020

Thank you very much for your positive response and recommendations!

The introduction will be extended and the discussion section will be added to address practical implications of the proposed method and other methods for lossy compression.

I like your idea of the diagram. It will be added. The plots will be refined to make them more readable.

---

## Author Response (AR1)

**Responses to the Interactive comments on "A note on precision-preserving compression of scientific data"**

Rostislav Kouznetsov[1,2]

[1]Finnish Meteorological Institute, Helsinki, Finland
[2]Obukhov Institute for Atmospheric Physics, Moscow, Russia

**Correspondence:** Rostislav Kouznetsov (Rostislav.Kouznetsov@fmi.fi)

I would like to thank all the reviewers again for their valuable comments The general points of the reviewers have been replied in corresponding short replies. More detailed and specific replies are below.

**1 Response to the Short comment by Milan Klöwer**

In addition to the response already given to the comment SC1, the following changes have been added.

5    *I suggest to discuss how your method is different from round-to-nearest tie-to-even which is already mentioned in IEEE-754 from 1985 as "An implementation of this standard shall provide round to nearest as the default rounding mode.*

**Response:** The description of integer-arithmetic rounding has been included together with an example of error accumulation due to half-to-infinity rounding method. Implementations of the standard half-to-even rounding both in Python and in high-level Fortran 95 are now included in the appendix.

**10  2 Response to the Reviewers comment by Mario Acosta**

. . .

*How should the rounding bit size would be decided? Is this a parameter so set up through NetCDF?*

**Response:** Conclusion has been extended to explicitly clarify that precision trimming can be used to facilitate NetCDF compression, while it is applicable to any format that stores floating-point data. Also a note on the precision control via NCO

15 has been added.

*Will the precision depend on the particular application or not? Apart from the array examples presented, the author should explain more in detail this.*

**Response:** The precision needed to store the data depends on both the data the intended application. A section with discussion on the precision needed for specific fields added.

20  *line 14: Consider 32 bits as float can be confused since this depends on the program language. Actually, some languages define float as double precision, using integer (int) for this declaration. If a more in detail explanation is not done, I would remove float and double*

**Response:** Removed

*lines 17-23: Have you consider different scientific fields to affirm that level of accuracy (less than 7 bits) or entropy? I agree*
25 *that in most cases single precision is more than enough but you should enumerate cases where this could not be true and provide a state of the art about this. As an example, I consider that in the inputs for data assimilation of chaotic applications, as weather models, the level of accuracy or entropy to ensure the number of bits used could affect the results*

**Response:** A section with real-world examples has been added. In particular, there are examples where a precision beyond the verifiable range is needed, and an example where virtually any precision would be sufficient, but the size is critical.
30 Concerning the inputs for the data assimilation, I would argue that if a small variation of the assimilation input (below the accuracy of the input data) significantly affects the results, there must be something wrong with the modelling setup.

*Introduction: I consider that the state of the art should be extended. In order to support the argument explained in the second paragraph, some successful examples should prove that the reduction of precision does not affect to the accuracy. Both of them should be listed, applications using the reduction of precision and the reduction of precision to save data.*
35 **Response:** The introduction was extended with summary of commonly-used lossy compression methods. Successful examples might be misleading, since there are also examples of the opposite. Instead a section with specific cases was added to illustrate that acceptable distortion strongly depend on the data and the applications.

*Introduction: Is precision-trimming the single compression method available? Apart from a default method, is it the most used or one of the best methods for Netcdf files? The author should explain why this particular method and why the comparison*
40 *done should be interesting for the reader, including some state of the art about this (apart from the single reference provided).*

**Response:** A description of Linear Packing, the most common to-date lossy compression method, added to the introduction, and a section illustrating drawbacks of LP added. Besides that a note on a modification of precision-trimming to keep absolute precision added.

*The mantissa-rounding technique explained here should highlight if it is novel or there are other works which present similar*
45 *approaches, or at least, differences among them.*

**Response:** I am not aware of any publications on mantissa-rounding, but I would be very surprised if no one used mantissa-rounding technique before. It has been implemented in the Silam model for several years already, without considering it worth a publication. The motivation for the current publication was the need to explore the features of precision-trimming, and the fact that a very popular software uses Bit Grooming that has no advantages over the rounding and introduces unnecessary
50 distortions.

*Section 2 should be rewritten. Second paragraph is difficult to follow and the explanation about the problem of trimming LSBs difficult to understand.*

**Response:** Section was modified. Hopefully the problem of trimming LSBs is more clear now. Besides that a simpler implementation of rounding is used, and an importance of half-to-even rounding highlighted.

55 *Five methods are described but only three of them are commented and only two discussed in details without more explanations about the reasons.*

**Response:** The reason has been explained and a short comment added.

*It is also not clear how these results in Figure 1 are obtained.*

**Response:** Figure 1 was generated by the script from Supplementary material. The note added to the "Code availability"

60 section. *Moreover, it is not clear if these results and the explanation given is coming from Zender (2016) paper or from Section 3 and 4 of this paper. The author should highlight the novel contribution of this paper for the different sections.*

**Response:** The paragraph re-phrased to more clearly distinguish the novel contribution.

*As minor details about the figures, X and Y axes for all of them do not contain units. Figure 2: a) and b) are difficult to see.*

**Response:** X and Y have same arbitrary units. Axis titles added, fonts increased.

65 ## 3 Response to the Reviewers comment by Seth McGinnis

. . . *The subject is a little dry; I think the paper would benefit from a brief discussion in the Introduction of the real-world use-case that led to the discovery of the two-point distortion, which would help the reader to follow the analysis using the structure function and understand why the issue matters.*

**Response:** A note on the stumbling on the issue of structure functions added to the introduction.

70 *The figure placement is a bit off. . . .*

**Response:** Should get better now.

*Figures 3 and 4 need axis labels. I would also recommend placing the legend outside the panels*

**Response:** The labels added. The legend shortened, so it does not overlap with curves.

**4 Response to the Reviewers comment by Ananda Kumar Das**

75 . . . *A little bit of description may be added in the introduction to establish the need of such a method. As the same has been mentioned by other reviewers, the elaboration on this point is skipped here.*

**Response:** A paragraph justifying the need for lossy compression added to the introduction.

*The methodology part of the paper which describes different precision algorithm based on the static specification of the bits of mantissa is lacking a little visibility without schematics through diagram. The diagram showing distribution of values in the*

80 *32 bits may be schematically presented for a broad class of readers.*

**Response:** A diagram added.

*If within the scope of the paper, the use of the specified method within NCO for real data may be tested and test result statistics may be added in the appendix.*

**Response:** Any example of a successful application of specific trimming parameters would allow for another realistic example when the same parameters were suboptimal for slightly different application or data, or even led to the data loss.

The intended topic of the paper is somewhat wider than the improvement and testing of NCO. For specific uses of NCO it is better to refer to the NCO documentation. While NCO has a means to control the precision, it can be done much finer: at low precisions every next tail-bit brings noticeable size reduction, which can be successfully utilized if a user clearly understands gains and losses of the precision-trimming, understands the needed precision, and has a fine-grain control over it. Therefore, the paper has been extended with illustrations of the features of precision-trimmed data including their compressibility, and with a discussion of needed precision and trimming methods for several real-world fields and applications.

I hope, it is a better alternative.